# SleepFM: Multi-modal Representation Learning for Sleep across ECG, EEG and Respiratory Signals

**Rahul Thapa[1], Bryan He[2], Magnus Ruud Kjær[3], Gauri Ganjoo[3], Hyatt Moore[3], Emmanuel Mignot[3], James Zou[1],**

[1] Department of Biomedical Data Science, Stanford School of Medicine, Stanford, CA, USA
[2] Department of Computer Science, Stanford School of Engineering, Stanford, CA, USA
[3] Department of Psychiatry and Behavioral Sciences, Stanford School of Medicine, Stanford, CA, USA
rthapa84@stanford.edu

## Abstract

Sleep is a complex physiological process involving multiple modalities across the body. We curate a large dataset of simultaneous polysomnography (PSG) recordings comprising electrical brain activity (EEG), heart rhythms (ECG), and respiratory patterns from over 14,000 participants, totaling over 100,000 hours of sleep data. We develop *SleepFM*, the first multi-modal foundation model for sleep learned through contrastive learning on this highly heterogeneous physiological data. When evaluated on a held-out test set, *SleepFM* significantly improves retrieval performance over 500x over random chance. A logistic regression model trained on *SleepFM*'s learned embeddings achieves strong performance on sleep stage classification (macro AUPRC 0.69) and apnea detection (AUPRC 0.71), outperforming an end-to-end trained CNN for sleep stage classification (AUPRC 0.579) and apnea detection (AUPRC 0.56). We find representations learned using an innovative leave-one-out approach during contrastive learning significantly improve downstream task performance compared to representations from standard pairwise contrastive learning. This work demonstrates the value of holistic multi-modal sleep modeling.

## Introduction

Sleep monitoring is a critical aspect for not only understanding sleep disorders but also gaining valuable insights into overall brain, pulmonary, and heart health (Worley 2018). Polysomnography (PSG), a comprehensive overnight sleep study, serves as a powerful tool by recording various physiological signals during sleep, including electroencephalogram (EEG), electrooculograms (EOG), and electrocardiogram (ECG) (Kryger, Roth, and Dement 2010). Traditionally, PSG data analysis involved manual visual inspection, a labor-intensive and time-consuming process prone to errors (Boashash and Ouelha 2016; Hassan and Bhuiyan 2017). Recent advancements in supervised deep learning have shown promise in automating sleep stage classification, particularly for disorders like apnea (Nassi et al. 2021; Olesen et al. 2021). However, most methods rely on labeled data from a narrow task. They rarely leverage the full breadth of physiological dynamics across diverse PSG modalities.

In parallel, contrastive learning has emerged as a powerful technique in other domains, such as radiology and pathology, where it pairs images with corresponding medical reports to learn rich medical image representations (Zhuang et al. 2022; Huang et al. 2021; Boecking et al. 2022; Bannur et al. 2023; Lu et al. 2023). However, PSG representation learning by pairing different channels via multi-modal contrastive learning has been less explored. While some uni-modal contrastive learning methods have been applied to ECG data (Kiyasseh, Zhu, and Clifton 2021; Diamant et al. 2022; Gopal et al. 2021; Mehari and Strodthoff 2022; Oh et al. 2022), they lack the ability to compare different modalities effectively in latent space, which is crucial for transfer learning. Additionally, (Raghu et al. 2022) developed SimCLR-like contrastive learning models pre-trained using multi-modal clinical time series data including ECG signals and structured time series data, and (Lalam et al. 2023) utilized a large collection of electronic health records (EHRs) to learn ECG representations through contrastive learning between ECG, structured and unstructured EHR data. However, these studies primarily focused on ECG data rather than the broader spectrum of PSG modalities investigated here.

**Our Contribution** We introduce *SleepFM*, a sleep foundation model trained using contrastive learning on a multi-modal PSG dataset comprising of 14K instances from a sleep study conducted at a major US academic hospital dating back to 1999. By capitalizing on EEG, ECG, and respiratory modalities from PSG, *SleepFM* exhibits superior performance in tasks such as retrieval, sleep stage classification, and apnea event classification, outperforming end-to-end trained CNN models. Additionally, our study highlights the potential of our methodology in scenarios with limited data availability, demonstrating promising results in a few-shot evaluation setting. To our knowledge, this is the first attempt to build and evaluate a foundation model for sleep.

## Method

### Datasets and Preprocessing

Our dataset encompasses PSG records from a US Sleep Clinic dating back to 1999. Comprising 14,068 recordings, this dataset features diverse waveforms, such as EEG, ECG, and EOG, collected over 8 hours per individual. All the

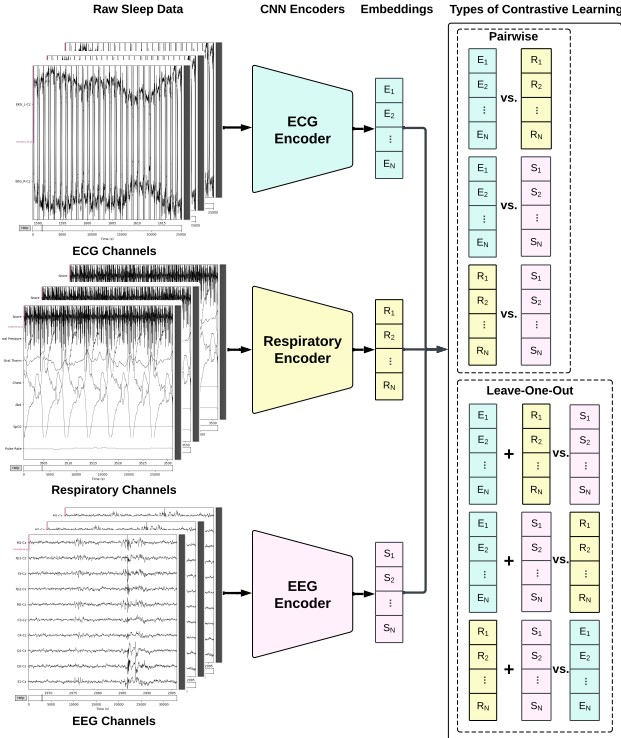

Raw Sleep Data | CNN Encoders | Embeddings | Types of Contrastive Learning

Figure 1: Overview of multi-modal PSG pre-training with contrastive learning.

data are deidentified to protect Protected Health Information (PHI) of the participants.

Cohort selection was based on three primary modalities: Respiratory, EEG, and ECG, encompassing 7, 10, and 2 channels, respectively. The specific channel names and descriptions are detailed in Table 6. The Respiratory modality includes channels measuring chest and abdomen movements, pulse readings, nasal and oral measurements, while EEG comprise electrodes gauging brain activity from various brain regions. ECG contains channels that measure cardiac function. Subsequently, we converted the total sleep duration into 30-second epochs for all participants and resampled the dataset to 256 Hz. Furthermore, we extracted labels corresponding to each epoch, as annotated by expert sleep technicians. Extracted labels include: Wake, Stage 1, Stage 2, Stage 3, REM, Sleep Apnea. To maintain data integrity and prevent leakage, the dataset is split into patient-level pretrain/train/validation/test sets: 11,261, 1,265, 141, and 1,401 participants respectively. Each patient contributes multiple clips to our dataset. The pretrain dataset is only ever used to pretrain our foundation model. The remaining set serves to train and test our model and baseline models for downstream applications. The validation set is used to optimize the hyperparameters. Demographic statistics for different splits are presented in Table 5. An illustrative snapshot of our data, accompanied by associated labels, can be found in Figures 3.

## Embedding Model

We used Convolutional Neural Network (CNN) to generate embedding from respiratory, EEG, and ECG channels. To handle the three distinct modalities of data, we developed three separate models. These models mainly differ in their input layers, which accommodate the number of channels specific to each modality: 10 for EEG, 2 for ECG, and 7 for respiratory channels. The architecture of these embedding models is rooted in the lightweight CNN architectures of MobileNetV2 and EfficientNet (Sandler et al. 2018; Tan and Le 2019). The architecture starts with atrous convolutions followed by subsequent multi-channel 1D convolutions. The layer count aligns with the original design of EfficientNet (Tan and Le 2019), but we significantly reduced the number of layers to less than 1/10th the size of the referenced architectures, aiming to optimize model runtime efficiency and minimize complexity. Following the initial atrous layers, our model incorporates convolutional layers utilizing a residual structure, mirroring the input and output bottleneck layers with an expansion layer (Sandler et al. 2018).

## Multi-modal Contrastive Learning

We explore two contrastive learning frameworks for learning joint representations: pairwise and leave-one-out contrastive learning (CL). The key idea is to bring positive pairs of embeddings from different modalities closer in the latent space while pushing apart negative pairs. The positive pairs are derived from either temporally aligned 30-second clips across modalities. All other non-matching instances within a training batch are treated as negative pairs.

In pairwise CL, we construct contrastive prediction tasks between all pairs of modalities. Specifically, for an embedding $x_i$ from modality $i$ and an embedding $x_j$ from modality $j$, we use a contrastive loss to encourage agreement between positive pairs while discouraging agreement between negative pairs. The contrastive prediction task is defined as:

$$l_{i,j,k} = -\log \frac{\exp(\text{sim}(x_{i,k}, x_{j,k}) * \tau)}{\sum_{m=1}^{N} \exp(\text{sim}(x_{i,k}, x_{j,m}) * \tau)} \quad (1)$$

where, N is the number of samples in a batch. This is the loss for a sample $k$ from modality $i$ in a given batch. We sum this loss over all the samples in a batch and repeat the process for all pairs of modalities $i, j$. The final loss is the sum of pairwise contrastive losses over all modality pairs.

In leave-one-out CL, we construct a predictive task where an embedding from one modality tries to identify the corresponding embeddings from the remaining modalities. In particular, for each modality $i$, we construct an embedding $\bar{x}_{\neq i}$ by averaging over embeddings from other modalities, excluding modality $i$. We apply a contrastive loss between modality $i$'s and this leave-one-out representation:

$$l_{i,j,k} = -\log \frac{\exp(\text{sim}(x_{i,k}, \bar{x}_{\neq i,k}) * \tau)}{\sum_{m=1}^{N} \exp(\text{sim}(x_{i,k}, \bar{x}_{\neq i,m}) * \tau)} \quad (2)$$

## Model Training

Our model pretraining, involves contrastive learning optimization with stochastic gradient descent (SGD) using a momentum of 0.9 and an initial learning rate set to 1e-2. We use cross-entropy as our loss function. Training spans 20 epochs with early stopping based on validation loss, employing a batch size of 32. Hyperparameters draw from similar models in prior literature (Ouyang et al. 2022).

Upon pretraining completion, we generate embeddings for the train, validation, and test sets, utilizing the learned modality encoders. These training embeddings drive the training of a logistic regression classifier. The classifier's performance undergoes evaluation on the test set for both sleep stage and apnea detection tasks. For comparison, we define a baseline EfficientNet architecture akin to our pre-trained model encoder but solely trained via supervised learning on the entire (pretraining + training) dataset for classification tasks. This model is trained end-to-end from scratch using cross-entropy loss between predicted and true labels, optimized by SGD with a step decay learning rate schedule. Mirroring the pretraining phase, this model undergoes training for 20 epochs with a batch size of 32, aligning hyperparameters with our model pretraining strategy. All model training was executed on a single NVIDIA Tesla V100S GPU with 32GB of memory. Pretraining each epoch consumed approximately 4 hours, while baseline supervised training required roughly 2 hours on the same GPU.

## Experiments

### Retrieval Analysis

We assessed our model's capabilities by retrieving one modality's closest embeddings from the test set based on another modality's embeddings. Computing cosine similarity between ECG and EEG embeddings generated a ranked list, allowing us to gauge retrieval performance. Evaluation was measured using recall@10 and median rank metrics. **Recall@10**: Measures the true paired item's appearance within the top 10 recommendations. Higher values indicate more accurate retrieval. **Median rank**: Determines the median position of the true paired item in rankings; a lower median rank signifies a more consistent ranking of the correct pair. We assessed the retrieval performance using 90,000 randomly selected 30-second clips encompassing all modalities from the test set. The baseline Recall@10 performance stands at $10/90000 = 0.0001$.

### Downstream Classification Tasks

We used the embeddings learned by the three models to train a logistic regression model. This model was employed to classify sleep stages and apnea events, and evaluation was performed on a held-out test dataset. Sleep stage classification is a multi-class classification task, with 5 classes: Wake, Stage 1, Stage 2, Stage 3, and REM. Apnea classification is a binary classification task. We compared our model's performance with baseline model, trained on all three modalities, for sleep stage and apnea event classification. Our evaluation relied on two primary metrics: AUROC (Area Under the

Table 1: Retrieval on the test set for model trained with pairwise contrastive learning. Resp is for Respiratory. Random baseline for Recall@10 = 0.0001

|      | Median Rank | | | Recall@10 | | |
|------|------|------|------|------|------|------|
|      | ECG | Resp | EEG | ECG | Resp | EEG |
| ECG  | -   | 2   | 1   | -    | 0.81 | 0.74 |
| Resp | 2   | -   | 5   | 0.82 | -    | 0.60 |
| EEG  | 1   | 6   | -   | 0.82 | 0.58 | -    |

Table 2: Retrieval on the test set for model trained with leave-one-out contrastive learning. Resp is for Respiratory. Random baseline for Recall@10 = 0.0001

|      | Median Rank | | | Recall@10 | | |
|------|------|------|------|------|------|------|
|      | ECG | Resp | EEG | ECG | Resp | EEG |
| ECG  | -   | 19  | 7   | -    | 0.39 | 0.58 |
| Resp | 21  | -   | 400 | 0.38 | -    | 0.05 |
| EEG  | 13  | 416 | -   | 0.46 | 0.05 | -    |

Receiver Operating Characteristic curve) and AUPRC (Area Under the Precision-Recall Curve).

### Few-Shot Evaluation

We performed few-shot evaluation by steadily increase the number of participants $k$ that each model sees from $k = 1$ to the full training dataset, and record the model's AUROC and AUPRC at each $k$. Note that each patient contributes multiple training clips. We consider values of $k \in \{1, 2, 4, 8, 16, 32, 64, 128, 1265\}$, where 1265 is the size of the full training set. For supervised CNN, few-shot examples are the only training examples seen by the model. For the pretrained models, we use embeddings of these few-shot examples to train a logistic regression model.

## Results

### Retrieval Analysis

Retrieval evaluation exhibited significant improvement compared to baseline metrics. Our model achieved over 500x-7000x higher recall@10 than the baseline as shown in Tables 2 and 1. Pairwise contrastive learning yield better overall retrieval performance than leave-one-out, most likely because the retrieval evaluation directly maps the training procedure of pairwise. One observable trend across both retrieval evaluation is relatively lower retrieval performance between Respiratory and other modalities, specially, Respiratory and EEG. The discrepancy in retrieval performance between EEG-Respiratory signals compared to EEG-ECG or ECG-Respiratory pairs might stem from the closer similarity and shared electrical nature between EEG and ECG signals. Both EEG and ECG capture electrical activities within the body, potentially resulting in more recognizable patterns and facilitating better correspondence.

Table 3: Sleep stage classification. Baseline here is an end-to-end CNN trained on the entire (pretraining + training) dataset to classify sleep stages. The leave-one-out and pairwise models are logistic regression models trained on the embeddings generated from only the training dataset. Prevalence of Wake, Stage 1, Stage 2, Stage 3, and REM are 0.21, 0.07, 0.51, 0.09, and 0.12 respectively. $\pm$ represents 95% CI.

| | AUROC | | | AUPRC | | |
|---|---|---|---|---|---|---|
| | **Leave-One-Out** | **Pairwise** | **Supervised CNN** | **Leave-One-Out** | **Pairwise** | **Supervised CNN** |
| Wake | $0.945_{\pm.001}$ | $0.930_{\pm.001}$ | $0.869_{\pm.001}$ | $0.862_{\pm.002}$ | $0.827_{\pm.002}$ | $0.711_{\pm.002}$ |
| Stage 1 | $0.814_{\pm.002}$ | $0.782_{\pm.002}$ | $0.706_{\pm.002}$ | $0.233_{\pm.003}$ | $0.186_{\pm.002}$ | $0.130_{\pm.002}$ |
| Stage 2 | $0.891_{\pm.001}$ | $0.861_{\pm.001}$ | $0.840_{\pm.001}$ | $0.876_{\pm.001}$ | $0.849_{\pm.001}$ | $0.822_{\pm.001}$ |
| Stage 3 | $0.928_{\pm.001}$ | $0.918_{\pm.001}$ | $0.918_{\pm.001}$ | $0.676_{\pm.003}$ | $0.615_{\pm.003}$ | $0.695_{\pm.002}$ |
| REM | $0.951_{\pm.001}$ | $0.891_{\pm.001}$ | $0.878_{\pm.001}$ | $0.778_{\pm.003}$ | $0.565_{\pm.002}$ | $0.540_{\pm.003}$ |
| **Avg** | **0.906** | 0.876 | 0.842 | **0.685** | 0.608 | 0.579 |

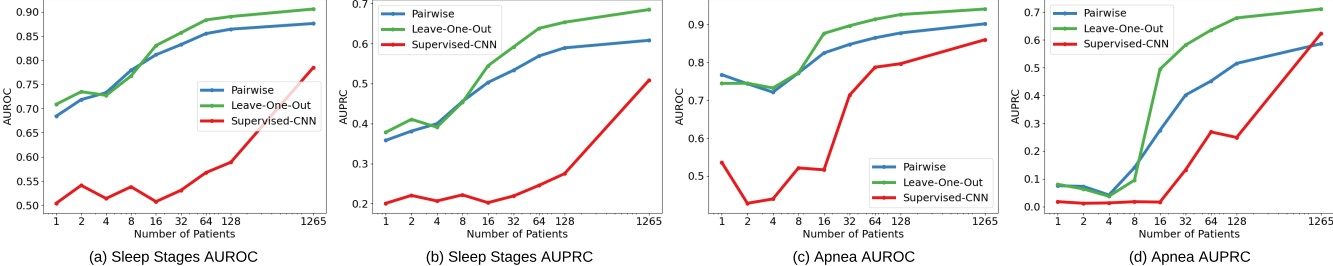

(a) Sleep Stages AUROC     (b) Sleep Stages AUPRC     (c) Apnea AUROC     (d) Apnea AUPRC

Figure 2: Sleep apnea classification. The x-axis represents number of participants that the model was trained on. In case of pairwise and leave-one-out, we select embeddings from $k$ number of participants to train a logistic regression model. In case of supervised CNN, we train the model end-to-end on $k$ number of participants to classify either sleep stages or apnea. Testing is done on the entire test set. For each shot, we average the performance across 3 replicates.

Table 4: Apnea classification metrics. Baseline here is a supervised CNN trained on the entire (pretraining + training) dataset to classify apnea. The leave-one-out and pairwise models are logistic regression models trained on the embeddings generated from only the training dataset. Prevalence of apnea event is 0.017. $\pm$ represents 95% CI.

| | AUROC | AUPRC |
|---|---|---|
| **Leave-One-Out CL** | $\mathbf{0.941}_{\pm.002}$ | $\mathbf{0.711}_{\pm.006}$ |
| **Pairwise CL** | $0.902_{\pm.003}$ | $0.586_{\pm.007}$ |
| **Supervised CNN** | $0.843_{\pm.002}$ | $0.555_{\pm.005}$ |

## Downstream Classification Tasks

We focused on two relevant sleep study tasks: sleep stage and apnea classification as shown in Table 3. Notably, across all metrics, the logistic regression model trained using representations from our pretrained model outperforms the end-to-end trained CNN. This superiority holds true across all sleep stage classes as well as on aggregated class metrics. Model pretrained with leave-one-out contrastive learning performs better than the one pretrained with pairwise contrastive learning. Similarly, the apnea classification metrics, displayed in Table 4, underscore our approach's superiority over supervised CNN models. Here as well, the model pretrained with leave-one-out contrastive learning significantly

outperforms the model pretrained with pairwise.

## Few-Shot Evaluation

The results for our few shot analysis is presented in Figure 2. We observe that across all the few shot settings, our model significantly outperforms baseline supervised CNN model for both sleep stage and apnea classification. Notably, the leave-one-out model significantly outperforms pairwise model across all shots, specially for apnea classification.

## Conclusion

Our study utilizes multi-modal PSG data and representation learning to improve identification of sleep events, advancing sleep medicine. The primary contributions involve developing and evaluating *SleepFM*, a multi-modal contrastive learning model, on a 14K PSG recordings. *SleepFM* exhibited strong performance across retrieval, sleep stage, and apnea classification, surpassing supervised CNNs. The methodology centers on two contrastive learning approaches, leave-one-out and pairwise, which both effectively unified ECG, EEG, and respiratory signal representations and demonstrated efficacy in limited data scenarios. For retrieval, pairwise contrastive learning outperformed leave-one-out. For all downstream tasks, leave-one-out significantly outperforms pairwise.

**Limitations.** We primarily trained and evaluated on one

dataset, thus model generalizability to other datasets is unknown. Testing on diverse datasets from different sleep clinics and demographics is crucial for validating robustness across populations. Additionally, while we focused on sleep stage and apnea detection, exploring other tasks like arousal detection, periodic leg movements, and narcolepsy could provide a more comprehensive clinical assessment.

# Appendix

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

Table 5: Demographics table. REM: Rapid Eye Movement; AHI: Apnea-Hypopnea Index, a measure used in sleep medicine to assess the severity of sleep apnea; WASO: Wake After Sleep Onset, the total time spent awake after initially falling asleep; SL: Sleep Latency, the time it takes to transition from wakefulness to sleep; REML: REM Sleep Latency, the time it takes to enter REM sleep after falling asleep; SE: Sleep Efficiency, the percentage of time spent asleep while in bed; TSD: Total Sleep Duration, the overall duration of sleep.

|  | pretrain | train | valid | test |
|---|---|---|---|---|
| participants (count) | 11,261 | 1,265 | 141 | 1,401 |
| events (count) | 10,611,314 | 1,190,392 | 130,380 | 1,314,267 |
| Duration (hours) | 88,427 | 9,920 | 1,086 | 10,952 |
| Male (%) | 49.85 | 50.15 | 47.12 | 53.04 |
| Female (%) | 43.83 | 43.99 | 48.08 | 41.79 |
| Unknown (%) | 6.32 | 5.86 | 4.8 | 5.17 |
| Age (years) | $42.19 \pm 19.63$ | $43.02 \pm 20.33$ | $40.41 \pm 19.98$ | $41.9 \pm 19.92$ |
| TSD (mins) | $376.78 \pm 90.84$ | $376.44 \pm 90.62$ | $371.22 \pm 84.9$ | $374.25 \pm 87.49$ |
| WASO (mins) | $79.4 \pm 60.54$ | $79.68 \pm 62.3$ | $78.76 \pm 57.27$ | $81.46 \pm 62.76$ |
| SE (mins) | $88.63 \pm 246.43$ | $91.93 \pm 91.68$ | $91.49 \pm 53.46$ | $92.36 \pm 64.17$ |
| SL (mins) | $22.16 \pm 32.75$ | $21.23 \pm 31.57$ | $28.99 \pm 87.76$ | $22.53 \pm 32.6$ |
| REML (mins) | $151.97 \pm 102.64$ | $149.41 \pm 97.72$ | $148.63 \pm 99.93$ | $154.87 \pm 103.53$ |
| Stage 1 (%) | $9.35 \pm 9.18$ | $9.31 \pm 8.75$ | $8.18 \pm 7.68$ | $9.04 \pm 8.86$ |
| Stage 2 (%) | $64.97 \pm 14.67$ | $64.79 \pm 14.72$ | $64.76 \pm 14.66$ | $64.97 \pm 14.72$ |
| Stage 3 (%) | $10.18 \pm 13.22$ | $10.2 \pm 13.19$ | $10.9 \pm 12.68$ | $10.32 \pm 13.57$ |
| REM (%) | $15.5 \pm 7.85$ | $15.7 \pm 8.01$ | $16.16 \pm 6.84$ | $15.67 \pm 7.88$ |
| AHI ($h^{-1}$) | $22.15 \pm 79.3$ | $22.77 \pm 19.14$ | $22.15 \pm 18.48$ | $20.89 \pm 16.96$ |

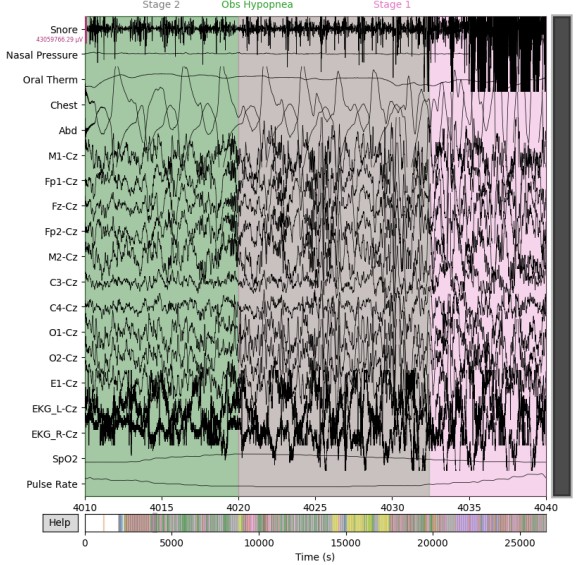

(a) 30-second clip of raw patient data. The x-axis is time and y-axis is different channels across all three modalities: EEG, ECG, and Respiratory.

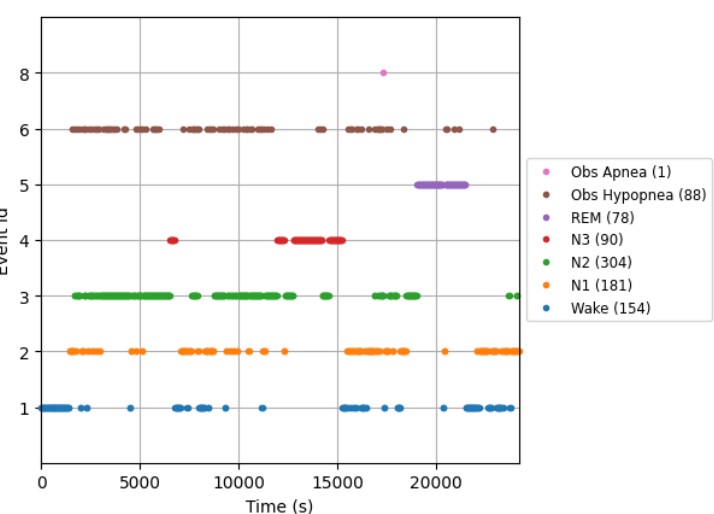

(b) Distribution of events across an entire patient sleep. The x-axis represents approximately 8 hours in seconds, and y-axis is distribution of different sleep events during the entire duration of sleep.

Figure 3: Raw signal data and corresponding events from a patient PSG.

Table 6: Data modalities, their associated channels and descriptions.

| Modality | Channel | Description |
|---|---|---|
| Respiratory | Chest | Measures expansion and effort to breathe. Vital in detecting sleep apnea and hypopneas. |
| | Snore | Detects vibrations or sound near airway openings during breathing. Identifies snoring patterns. |
| | SpO2 | Measures blood oxygen saturation using a clip on the fingertip or earlobe. Important for identifying variations in oxygen levels. |
| | Abdomen | Measures expansion and effort to breathe. Complements the chest belt in detecting respiratory efforts. |
| | Pulse Rate | Calculated from fingertip or ECG signals. Indicates respiratory disturbances. |
| | Nasal Pressure | Detects airflow limitations and obstructions, aiding in identifying nasal breathing difficulties. |
| | Oral Therm | Assesses nasal/oral breathing temperature. Detects mouth breathing affecting sleep quality. |
| Sleep Stages | E1 | Electrooculogram near the left eye, monitoring eye movements for sleep stages. |
| | M1 | Electromyogram on chin muscles. Monitors muscle activity for sleep cycles. |
| | M2 | Monitors chin muscle activity, aiding in differentiating REM and NREM sleep. |
| | C3 | EEG on the left hemisphere. Captures brainwave patterns for sleep staging. |
| | C4 | EEG on the right hemisphere. Captures brainwave patterns for sleep staging. |
| | O1 | EEG on the back left of the head. Captures brainwave patterns during NREM sleep. |
| | O2 | EEG on the back right of the head. Captures brainwave patterns during NREM sleep. |
| | Fz | Frontal EEG on the forehead. Captures brainwave patterns related to cognitive processes. |
| | Fp1 | Prefrontal EEG on the left forehead. Monitors brainwave patterns related to emotional processing. |
| | Fp2 | Prefrontal EEG on the right forehead. Monitors brainwave patterns related to emotional processing. |
| ECG | ECG_L | Left ECG electrode measures heart's electrical activity, aiding in detecting arrhythmias. |
| | ECG_R | Right ECG electrode monitors heart's electrical activity to identify arrhythmias. |

Table 7: Sleep stage classification metrics for model trained with leave-one-out contrastive learning. After having trained the model with all three modalities, we extract embeddings for each modality separately and train a logistic regression with each modality to identify sleep stages. $\pm$ represents 95% confidence intervals.

| | AUROC | | | AUPRC | | |
|---|---|---|---|---|---|---|
| | ECG | Respiratory | EEG | ECG | Respiratory | EEG |
| Wake | $0.934_{\pm.001}$ | $0.846_{\pm.001}$ | $0.942_{\pm.001}$ | $0.829_{\pm.004}$ | $0.652_{\pm.003}$ | $0.857_{\pm.002}$ |
| Stage 1 | $0.786_{\pm.002}$ | $0.676_{\pm.002}$ | $0.801_{\pm.002}$ | $0.193_{\pm.002}$ | $0.127_{\pm.001}$ | $0.211_{\pm.003}$ |
| Stage 2 | $0.874_{\pm.001}$ | $0.728_{\pm.001}$ | $0.888_{\pm.001}$ | $0.860_{\pm.001}$ | $0.708_{\pm.001}$ | $0.873_{\pm.001}$ |
| Stage 3 | $0.919_{\pm.001}$ | $0.788_{\pm.001}$ | $0.927_{\pm.001}$ | $0.638_{\pm.003}$ | $0.307_{\pm.002}$ | $0.679_{\pm.002}$ |
| REM | $0.939_{\pm.001}$ | $0.789_{\pm.001}$ | $0.944_{\pm.001}$ | $0.745_{\pm.003}$ | $0.388_{\pm.003}$ | $0.724_{\pm.003}$ |
| **Macro Avg** | 0.891 | 0.765 | 0.900 | 0.436 | 0.484 | 0.669 |

Table 8: Apnea classification metrics for model trained with leave-one-out contrastive learning. After having trained the model with all three modalities, we extract embeddings for each modality separately and train a logistic regression with each modality to identify apnea. $\pm$ represents 95% confidence intervals.

|  | ECG | Respiratory | EEG |
|---|---|---|---|
| AUROC | $0.735_{\pm.004}$ | $0.925_{\pm.002}$ | $0.735_{\pm.004}$ |
| AUPRC | $0.040_{\pm.001}$ | $0.697_{\pm.006}$ | $0.040_{\pm.001}$ |

Table 9: Sleep stage classification metrics for model trained with pairwise contrastive learning. After having trained the model with all three modalities, we extract embeddings for each modality separately and train a logistic regression with each modality to identify sleep stages. $\pm$ represents 95% confidence intervals.

|  | AUROC | | | AUPRC | | |
|---|---|---|---|---|---|---|
|  | ECG | Respiratory | EEG | ECG | Respiratory | EEG |
| Wake | $0.940_{\pm.001}$ | $0.877_{\pm.001}$ | $0.945_{\pm.001}$ | $0.838_{\pm.002}$ | $0.710_{\pm.002}$ | $0.866_{\pm.001}$ |
| Stage 1 | $0.791_{\pm.002}$ | $0.701_{\pm.002}$ | $0.812_{\pm.002}$ | $0.199_{\pm.002}$ | $0.140_{\pm.001}$ | $0.225_{\pm.002}$ |
| Stage 2 | $0.876_{\pm.001}$ | $0.760_{\pm.001}$ | $0.891_{\pm.001}$ | $0.862_{\pm.001}$ | $0.737_{\pm.001}$ | $0.872_{\pm.001}$ |
| Stage 3 | $0.917_{\pm.001}$ | $0.806_{\pm.001}$ | $0.925_{\pm.001}$ | $0.627_{\pm.002}$ | $0.339_{\pm.003}$ | $0.645_{\pm.003}$ |
| REM | $0.939_{\pm.001}$ | $0.839_{\pm.001}$ | $0.953_{\pm.001}$ | $0.761_{\pm.003}$ | $0.499_{\pm.003}$ | $0.797_{\pm.002}$ |
| **Macro Avg** | 0.892 | 0.796 | 0.905 | 0.657 | 0.484 | 0.680 |

Table 10: Apnea classification metrics for model trained with pairwise contrastive learning. After having trained the model with all three modalities, we extract embeddings for each modality separately and train a logistic regression with each modality to identify apnea. $\pm$ represents 95% confidence intervals.

|  | ECG | Respiratory | EEG |
|---|---|---|---|
| AUROC | $0.750_{\pm.003}$ | $0.916_{\pm.003}$ | $0.733_{\pm.004}$ |
| AUPRC | $0.041_{\pm.001}$ | $0.456_{\pm.006}$ | $0.036_{\pm.001}$ |