# OpenReview forum: "SleepFM: Multi-modal Representation Learning for Sleep across ECG, EEG and Respiratory Signals"
_AAAI.org/2024/Spring_Symposium_Series/Clinical_FMs — AAAI 2024 SSS on Clinical FMs_

### Official Review · Reviewer_82eE · 2024-02-21
**Promising Idea and Approach**

**Rating:** 5
**Confidence:** 4

**Review:**

### Summary

This paper proposes a foundational model for sleep-related tasks and shows its efficacy through two downstream tasks - (i) sleep apnea detection, and (ii) sleep stage detection. While it proposes a novel application of two interesting pre-training techniques, the experimentation does not truly evaluate the potential of the proposed foundational model. Overall, the paper is built on a promising idea and its presentation can be further improved through more rigorous experimentation.

### Strengths
- Novel application of pretraining tasks to align multiple modalities to create a sleep (time-series) foundational model
- Paper is written with clarity, explaining proposed methods and experiments well
- Promising results

### Weaknesses
- “Foundational Model” capabilities have not been evaluated: To claim that a new model is a foundational model, it must satisfy certain properties of foundational models (FM). It is the authors’ responsibility to then highlight the properties that have been tested and note limitations of the foundational model. For example, a common property found in foundational models is the ability to generalize reasonably well to an unseen dataset.
- Weak baselines: Ideally, the model should be compared with other methods which have been trained for the proposed downstream tasks. Eg: [1] is a sleep apnea detection method which can be included as a baseline.

### Other Feedback (to extend this work):
- The authors should motivate whether a foundational model for sleep is in fact useful. Identifying the shortcomings of existing methods (which do not consider all modalities, for example) and clearly delineating the research questions that you would like to answer would be a good way to approach this.
- Authors considered separate encoders for the different modalities. Some recent papers on time-series foundational modeling have shown that a single time-series model can encode time-series of different #channels and frequencies [2, 3]. Consider using one of these methods as the base model for your proposed pre-training tasks. This could help better align the modalities, especially if one of the modalities is more sparse than others.
- Experiment on the effect of leaving out one modality - could motivate the need to jointly model multiple modalities.

[1] https://ieeexplore.ieee.org/abstract/document/8571271
[2] https://arxiv.org/pdf/2302.11939.pdf
[3] https://arxiv.org/abs/2402.03885

---

### Official Review · Reviewer_Td6r · 2024-02-22
**Review and Suggestions for Improvement on a Novel Multi-Modal Sleep Event Identification Method Using Contrastive Learning**

**Rating:** 9
**Confidence:** 4

**Review:**

Reviewer's Comments:

The paper introduces an innovative method for sleep event identification based on multi-modal contrastive learning, which is innovative in the field of sleep medicine. The SleepFM model performs well in retrieval, sleep stage, and apnea classification, contributing to sleep medicine research. Here are some suggestions:

Language: The overall language of the paper is fluent, but there are instances where expressions are not clear. Additionally, there are numerous abbreviations without providing the original full terms. Particularly, for "SleepFM," I am curious about what "FM" stands for.

Methodology: The paper describes clear methods, including dataset selection, model architecture, and evaluation metrics. However, more detailed descriptions of the implementation details of the contrastive learning method may be needed for readers to understand the model training process and data processing flow.

Results and Discussion: The Results section provides detailed experimental results, but some explanations for certain results could be more in-depth. For example, why does pairwise contrastive learning perform better in retrieval? The authors could provide more explanations about the internal mechanisms of the model.

Overall, this paper is helpful for research in the field of sleep medicine, but there are still some aspects that can be further improved and refined.

---

### Official Review · Reviewer_rUdU · 2024-02-22
**Interesting work on Sleep foundation model**

**Rating:** 8
**Confidence:** 5

**Review:**

The paper presents "SleepFM," a multi-modal foundation model for sleep analysis using large-scale polysomnography (PSG) data, including EEG, ECG, and respiratory signals. The model is trained via contrastive learning and demonstrates superior performance in tasks like sleep stage classification and apnea detection compared to traditional CNN methods. The innovative aspect of using a leave-one-out approach in contrastive learning for multi-modal data integration is highlighted, showing significant improvements in model performance.

As author stated in the limitation section, more external datasets are needed to prove the work as a foundation model. Also, it would be great to see the performance of using only one kind of signal as downstream task, e.g. ECG-only for sleep staging.

---

### Official Review · Reviewer_w78d · 2024-02-22

**Rating:** 7
**Confidence:** 4

**Review:**

**Summary:**

The paper proposes SleepFM, a foundation model for sleep, trained using contrastive learning on a self-curated dataset consisting of EEG, ECG, and respiratory signals. The authors evaluate their model on downstream tasks like sleep stage classification and apnea event classification, showing performance superior to end-to-end trained CNN models. They also perform retrieval tests, showcasing their model’s ability to retrieve one modality’s closest embeddings from the test set based on another modality’s embeddings. The embedding layer in their model consists of three CNN encoders for each type of signal data, and the model is pretrained on a contrastive learning objective. The paper also evaluates the impact of pairwise contrastive learning vs leave-one-out contrastive learning objective, showing better performance on the downstream tasks using the latter. For classification, the model uses the embeddings from the pretrained model and uses them to train a logistic regression classifier to evaluate downstream performance. The model shows good performance compared to the baseline in the paper across a wide set of experiments.

**Strengths:**

1. The authors introduce a fairly large-scale multi-sensory dataset of simultaneous measurements of EEG, ECG, and EOG signals, focused on training a foundation model from scratch. This dataset seems well-curated, and based on the downstream results, the embeddings from the pre-trained model have a positive impact on the performance on given tasks.

2. The motivation for using a contrastive-learning-based pre-training methodology using all three types of time-series data is sufficiently articulated and well-grounded with prior work in the paper.

3. The pre-training, fine-tuning, validation, and test splits for the dataset are well-defined, mitigating the risk of any data contamination in the evaluation pipeline.

4. Owing to the availability of various types of paired time-series data in the pre-training dataset, the comparison of pairwise contrastive learning vs leave-one-out contrastive learning as the training objective was relevant and interesting.

5. The k-shot analysis of the model’s performance for classification was relevant, explicitly showcasing that contextual information learned during pretraining can improve downstream performance.

**Weaknesses:**

1. I am not sure I would call this model a “multi-modal” model, since the model is primarily trained on paired multi-variate time-series across different domains. EEG, ECG, and EOG data all lie in the time-series modality and are similarly modeled using the same type of encoders.

2. Experimental comparison with other recent statistical and deep learning methods for the given downstream tasks is necessary to have a more holistic understanding of the proposed model’s performance.

**Other recommendations:**

1. In future work, interpretability experiments to show what the model in learning would be interesting, and evaluating the model potentially in zero-shot settings would be appreciated as well.

2. Leave-one-out contrastive learning is not a new approach, and prior works from other domains [1,2] should be cited for the same.

    [1] Sanchez-Fernandez, A., Rumetshofer, E., Hochreiter, S. et al. CLOOME: contrastive learning unlocks bioimaging databases for queries with chemical structures. Nat Commun 14, 7339 (2023). https://doi.org/10.1038/s41467-023-42328-w

    [2] Xiao, T., Wang, X., Efros, A. A., & Darrell, T. (2021). What Should Not Be Contrastive in Contrastive Learning. International Conference on Learning Representations. https://openreview.net/forum?id=CZ8Y3NzuVzO